# Genome-Wide Identification, Characterization and Expression Analysis of the *TaDUF724* Gene Family in Wheat (*Triticum aestivum*)

**DOI:** 10.3390/ijms241814248

**Published:** 2023-09-18

**Authors:** Yi Yuan, Xiaohui Yin, Xiaowen Han, Shuo Han, Yiting Li, Dongfang Ma, Zhengwu Fang, Junliang Yin, Shuangjun Gong

**Affiliations:** 1Hubei Key Laboratory of Waterlogging Disaster and Agricultural Use of Wetland/Ministry of Agriculture and Rural Affairs (MARA) Key Laboratory of Sustainable Crop Production in the Middle Reaches of the Yangtze River (Co-Construction by Ministry and Province), College of Agriculture, Yangtze University, Jingzhou 434025, China; 2Key Laboratory of Integrated Pest Management of Crops in Central China, Ministry of Agriculture/Hubei Key Laboratory of Crop Diseases, Insect Pests and Weeds Control, Institute of Plant Protection and Soil Science, Hubei Academy of Agricultural Sciences, Wuhan 430064, China

**Keywords:** bioinformatics, *cis*-element, expression profiling, gene structure, miRNA, RT-qPCR, subcellular localization

## Abstract

Unknown functional domain (DUF) proteins constitute a large number of functionally uncharacterized protein families in eukaryotes. DUF724s play crucial roles in plants. However, the insight understanding of wheat *TaDUF724s* is currently lacking. To explore the possible function of *TaDUF724s* in wheat growth and development and stress response, the family members were systematically identified and characterized. In total, 14 *TaDUF724s* were detected from a wheat reference genome; they are unevenly distributed across the 11 chromosomes, and, according to chromosome location, they were named *TaDUF724-1* to *TaDUF724-14*. Evolution analysis revealed that *TaDUF724s* were under negative selection, and fragment replication was the main reason for family expansion. All TaDUF724s are unstable proteins; most TaDUF724s are acidic and hydrophilic. They were predicted to be located in the nucleus and chloroplast. The promoter regions of *TaDUF724s* were enriched with the *cis*-elements functionally associated with growth and development, as well as being hormone-responsive. Expression profiling showed that *TaDUF724-9* was highly expressed in seedings, roots, leaves, stems, spikes and grains, and strongly expressed throughout the whole growth period. The 12 *TaDUF724* were post-transcription regulated by 12 wheat MicroRNA (miRNA) through cleavage and translation. RT-qPCR showed that six *TaDUF724s* were regulated by biological and abiotic stresses. Conclusively, *TaDUF724s* were systematically analyzed using bioinformatics methods, which laid a theoretical foundation for clarifying the function of *TaDUF724s* in wheat.

## 1. Introduction

Wheat (*Triticum aestivum*) belongs to the Gramineae family, and has high nutritional value [1]. Its yield accounts for 35% of the world’s total grain production, ranking the third in the world in terms of output, second only to rice and corn [2]. However, wheat is affected by a variety of biotic and abiotic stresses during its growth and development, such as high temperature, drought, salinity, pests and diseases, which seriously affect the quality and yield of wheat [3]. Theory and practice have proved that breeding resistance cultivars are the most economical, environmentally friendly and effective strategy to control the biotic and abiotic stresses of wheat [4]. Therefore, it is very important to explore more genes with stress-resistance roles, to prepare gene resources for wheat resistance breeding [5].

The domains of unknown function (DUF) family refers to a series of protein families whose functions have not yet been annotated [6]. Currently, the Pfam database (V35.0) contains 4795 DUF families, accounting for 24% of all Pfam families [7]. Studies showed that different DUF proteins have a variety of functions. For example, in wheat, compared with control, the *TaDUF966-9B* knockdown plants exhibited severe leaf curling post inoculation with BSMV (barley stripe mosaic virus) under salt stress, suggesting that *TaDUF966* plays a vital role in salt stress tolerance in wheat [8]. In the study of rice *OsDUF668* expression profiling, all genes were responsive to drought stress, and some of the Avr9/Cf-9 rapidly elicited genes resist salt, wound and rice blasts with rapidly altered expression patterns. These results showed that *OsDUF668* played an important role in drought resistance and plant defense [9]. In Arabidopsis, the RNA transcription level of the *ATDUF4228* gene was significantly increased under osmosis, cold and salt treatment. The results indicated that *ATDUF4228* plays a potential role in plant stress response [10].

DUF724 is one family in the DUF superfamily, which contains highly conserved DUF724, Agenet and Tudor domains [11]. Furthermore, in plants, the EMSY-like N-Terminal (ENT), BAH, Plant Homeodomain (PHD) and DUF724 domains often co-occur with Agenet/Tudor domains, which leads to functional differentiation of the DUF724 protein [12]. For instance, SWO1 (SWOLLEN 1), a protein containing the Agenet domain, plays a role in salt stress response by interacting with importin α [13]. FMRP, a DUF724 domain-containing protein, can be involved in mediating protein–protein interactions, thus responding to DNA damage [14]. In Arabidopsis, the expression pattern of *DUF724* in different tissues was studied using RT-qPCR. The results indicate that most members of *DUF724* showed high expression levels in seedlings and flowers; some DUF724s exhibited extreme tissue-specificity patterns. For example, *AtDuf9* can be detected in flowers, whereas it is virtually undetectable in other organs. Meanwhile, family members clustered in the same phylogenetic branch showed similar expression patterns. In addition, the expression pattern of *DUF724* in *A. thaliana* was also studied using the GUS method. Almost all genes can be detected in plant tissues, including roots, stems, flowers, leaves and fruit; the results indicated that *DUF724* could play important roles in plant growth [15].

Although *DUF724* has been studied in Arabidopsis, it has not been explored in wheat. In this study, we conducted genome-wide identification and systematic analysis of *DUF724* genes in wheat, including phylogenetic analysis, chromosome localization, evolution analysis, gene structure and conserved Motif analysis, protein 3D model prediction, *cis*-element analysis, expression profiling and miRNA analysis. Meanwhile, the expression patterns of six *TaDUF724* genes were verified using RT-qPCR. This study will improve our understanding of *TaDUF724* genes in wheat, and lay a theoretical foundation for the functional study of genes.

## 2. Results

### 2.1. Identification and Phylogenetic Analysis of TaDUF724s

In this study, the Hidden Markov Model (HMM) seed sequence of DUF724 (PF05266) and protein sequences of Arabidopsis and rice DUF724s were used as query sequences to BLASTp search against the wheat genome database. Finally, 14 *TaDUF724* (*Triticum aestivum DUF724*) genes were identified from the reference genome. These *TaDUF724* genes are located on 11 wheat chromosomes (Table 1), and, based on their position on the chromosome, they were named *TaDUF724-1* to *TaDUF724-14*.

To further analyze the evolutionary relationships, a total of 29 DUF724 proteins were utilized for the construction of the phylogenetic tree (Figure 1). These proteins included ten Arabidopsis, five rice and fourteen wheat DUF724s. According to their evolutionary distance, they were divided into two groups: Group I and Group II. Group I contained wheat, rice and Arabidopsis DUF724s, whereas Group II contained only wheat and rice DUF724s. Group I contained three TaDUF724s, and Group II contained 11 TaDUF724s. Furthermore, wheat and rice DUF724s were clustered into close branches. The results showed that wheat and rice DUF724 family members had a close genetic evolutionary relationship.

### 2.2. Chromosome Localization, Evolution Analysis of TaDUF724s

The 14 *TaDUF724* genes identified were distributed on 11 chromosomes (Figure 2A). *TaDUF724-1* and *TaDUF724-2* were located on chromosome 2A; *TaDUF724-3* and *TaDUF724-4* were located on chromosome 2B; *TaDUF724-5* and *TaDUF724-6* were located on 2D chromosomes; and *TaDUF724-7, TaDUF724-8, TaDUF724-9, TaDUF724-10, TaDUF724-11, TaDUF724-12, TaDUF724-13* and *TaDUF724-14* were, respectively, located on chromosomes 3D, 4A, 4B, 5A, 5B, 6A, 6B and 6D. Two *TaDUF724s* were, respectively, located on 2A, 2B and 2D, and other chromosomes contained only one gene.

The results of collinear analysis (Figure 2B) show that homologous gene pairs were detected among chromosome 2A, 2B and 2D, and tandem replication occurs between *TaDUF724-1* and *TaDUF724-2, TaDUF724-3* and *TaDUF724-4*, and *TaDUF724-5* and *TaDUF724-6*. There were homologous gene pairs between chromosome 5A and 5B; fragment replication occurred between *TaDUF724-10* and *TaDUF724-11*. There were homologous gene pairs that were detected among chromosome 6A, 6B and 6D, and fragment replication occurred between *TaDUF724-12*, *TaDUF724-13* and *TaDUF724-14*. Using BLASTn analysis, 11 pairs of homologous genes were identified between wheat and its ancestors (*T. aestivum* vs. *Triticum urartu*, 1 pair; *T. aestivum* vs. *Triticum turgidum*, 6 pairs; *T. aestivum* vs. *T. aestivum*, 4 pairs) (Figure 2C), and the Ka/Ks values between all homologous gene pairs were less than one, suggesting that these genes evolved under the effect of negative selection.

### 2.3. Gene Structure and Conserved Motif Analysis of TaDUF724s

The phylogenetic tree and gene structure analysis results are shown in Figure 3A,B. All members of Group I contain introns and exons, and the number of introns ranges from three to seven; the number of exons ranges from four to eight; and the maximum number of introns and exons in *TaDUF724-9* is seven and eight. The *TaDUF724* genes in Group II contain only one exon, *TaDUF724-7* and *TaDUF724-10* have no intron and *TaDUF724-11* has only one intron.

The results of the conservative Motif analysis show that Motif1 and Motif9 are present in all the genes (Figure 3C). The genes in Group I contain six to ten Motifs; Group II genes contain only two Motifs. The number and structure of Motifs in the same subgroup are very similar; all the genes in Group I contain Motif2, Motif4 and Motif7, while those in Group II do not. Domain analysis found that Motif1 constituted the DUF724 domain, Motif6 constituted the Agenet domain and Motif7 and Motif8 constituted the Tudor_SF domain. Motif1, Motif6, Motif7 and Motif8 constitute the key functional domains of DUF724, and the other Motifs do not match known key functional domains.

### 2.4. Analysis of Protein Properties of TaDUF724s

Analysis of physical and chemical properties of proteins showed that TaDUF724s varied widely in amino acid number, with an average of 809 aa and amino acid lengths ranging from 523 to 980 aa (Table 1). The mean molecular weight was 89.9 kDa and the range was 57.8 (TaDUF724-3) to 108.9 (TaDUF724-9) kDa. The isoelectric point of TaDUF724 proteins ranged from 4.7 to 7.6, with an average value of 5.9 and less than 6. Therefore, most TaDUF724s were acidic proteins. The instability index of TaDUF724s ranged from 50.3 to 65, all of which were greater than 40, so all TaDUF724s were unstable proteins. The total average hydrophilicity of TaDUF724s ranged from −0.354 to −0.804, with an average value of −0.51, indicating that TaDUF724s had high hydrophilicity. Subcellular localization prediction showed that TaDUF724s were mainly distributed in the nucleus and chloroplast.

### 2.5. 3D-Structure Analysis of TaDUF724 Protein

The secondary structure of TaDUF724s was predicted using SOPMA (Table 2). The results show that TaDUF724s have four structural forms: α-helix, extended chain, β-angle and random curl. Among them, α-helix (32.45–44.81%) and random curl (39.01–49.64%) are the main types of protein secondary structures; the elongation chain (9.18~15.95%) and β-angle (5.02~7.27%) were smaller types of protein secondary structures.

Three-dimensional model construction of TaDUF724s was performed through the SWISS-MODEL online tool. The construction results showed that (Figure 4) TaDUF724-7, TaDUF724-10 and TaDUF724-11 in Group II have very simple 3D structures, which corresponded to the prediction results of protein secondary structures. As can be saw in Figure 4, the Group I proteins exhibited more complex three-dimensional structural features.

### 2.6. Promoter Cis-Element Analysis of TaDUF724s

The promoter *cis*-elements of 14 *DUF724* genes were studied; the upstream 1500 bp sequences of *TaDUF724* genes were analyzed using PlantCare. A total of 46 *cis*-elements were identified (Figure 5), including 18 light responsive elements, eight plant hormones, five stress-related elements and 15 plant growth elements. In the promoter sequence of each *TaDUF724* gene, CAAT-box and TATA-box are widely distributed and abundant, which is usually associated with the growth and development of plants. In addition, *cis*-acting regulatory elements involved in MeJA-responsiveness were also discovered in each gene (CGTCA-motif and TGACG-motif). The *cis*-acting elements involved in the abscisic acid responsiveness (ABRE), light responsiveness (G-box) and the *cis*-acting regulatory element essential for anaerobic induction (ARE) were commonly detected in *TaDUF724* promoter regions.

### 2.7. Analysis of TaDUF724 Gene Family Expression Pattern

In order to explore the potential role of *TaDUF724* genes, we used RNA-seq data to study the expression pattern of the *DUF724* gene in different developmental stages and tissues of wheat and under different stresses (Figure 6); the results showed that *TaDUF724-9* and *TaDUF724-10* are expressed at higher levels than other genes under plant growth and development, biological stress and abiotic stress conditions. *TaDUF724-8*, *TaDUF724-10* and *TaDUF724-11* were only expressed during growth and development. During growth and development, *TaDUF724-9* was highly expressed in all periods and tissues. The expression of *TaDUF724* under different abiotic stress conditions was analyzed. Drought treatment and PEG6000 treatment decreased the expression of *TaDUF724-8, TaDUF724-9*, *TaDUF724-10* and *TaDUF724-11* in leaves. *TaDUF724-9* and *TaDUF724-10* showed similar expression patterns under drought and PEG600 abiotic stresses. The remaining two genes were hardly or lowly expressed. Under biological stress, the expressions of *TaDUF724-9* and *TaDUF724-10* in wheat leaves decreased after infection by *Zymoseptoria tritici*. The expression levels of *TaDUF724-9* and *TaDUF724-10* in wheat leaf were increased after the infection of stripe rust. In addition, the expression levels of other genes under biological stress were not significantly different from those under normal conditions.

### 2.8. Post-Transcriptional Regulation of TaDUF724s by miRNA

As shown in Figure 7, a total of nine wheat miRNAs (tae-miR1120b-3p, tae-miR5050, tae-miR9653a-3p, tae-miR9653b, tae-miR9655-3p, tae-miR9667-5p, tae-miR9676-5p, tae-miR9773, tae-miR9780) target twelve *TaDUF724* genes through cleavage (*TaDUF724-1*, *TaDUF724-2*, *TaDUF724-3*, *TaDUF724-4*, *TaDUF724-6*, *TaDUF724-8*, *TaDUF724-9*, *TaDUF724-10*, *TaDUF724-11*, *TaDUF724-12*, *TaDUF724-13*, *TaDUF724-14*), and four wheat miRNAs (tae-miR9674a-5p, tae-miR5050, tae-miR9677a, tae-miR9677b) target five *TaDUF724* genes through translation (*TaDUF724-2*, *TaDUF724-3*, *TaDUF724-12*, *TaDUF724-13*, *TaDUF724-14*). As showed in Figure 7, among them, tae-miR5050 has both cleavage and translation inhibition effects for *TaDUF724-13*. The results showed that multiple miRNAs targeted *TaDUF724s* through cleavage and translation, which were involved in the post-transcriptional regulation of *TaDUF724s.*

### 2.9. Quantitative Real-Time PCR and Data Analysis

To further verify the specific expression profile of the *DUF724* gene in response to stress in wheat, we performed RT-qPCR experiment analysis of six highly expressed genes (*TaDUF724-8*, *TaDUF724-9*, *TaDUF724-10*, *TaDUF724-11*, *TaDUF724-12* and *TaDUF724-14*) (Figure 8) which have obvious responses to different biological and abiotic stresses.

Under PEG-induced drought stress, compared with the control, the expression levels of *TaDUF724-8*, *TaDUF724-11* and *TaDUF724-12* were up-regulated at all time points, and the expression levels of *TaDUF724-9*, *TaDUF724-10* and *TaDUF724-14* were fluctuation

Under the condition of NaCl stress, the expression level of *TaDUF724-8* was inhibited. The expression level of *TaDUF724-9* and *TaDUF724-12* was significantly up-regulated at all time points. The expression level of *TaDUF724-10 and TaDUF724-11* increased significantly after 2 h; however, the expression of *TaDUF724-11* decreased at 72 h.

Under the infection of *Fusarium graminearum*, the expression levels of all genes were highly induced at 6 h. After 6 h, the expression levels of *TaDUF724-8* and *TaDUF724-10* were slightly increased, and *TaDUF724-11*, *TaDUF724-12* and *TaDUF724-14* were significantly inhibited.

Under the infection of *Puccinia striiformis*, the expression levels of *TaDUF724-11*, *TaDUF724-12* and *TaDUF724-14* were significantly lower than those of the control, but the expression levels of *TaDUF724-12* and *TaDUF724-14* were gradually increased with the extension of infection time.

## 3. Discussion

In this study, we conducted a comprehensive genome-wide identification and expression pattern analysis of *TaDUF724s*. A total of 14 *TaDUF724* family members were identified in wheat (Table 1). *DUF724* genes encoding proteins with the DUF724 domain (conserved domain of unknown function 724) and Agenet domains were found in wheat and other plants, but not in animals and fungi [15]. High levels of *AtDuf724-4* promoter activity were detected in the petal tissue of Arabidopsis. In stems, the expression of *AtDuf724-1*, *AtDuf724-5*, *AtDuf724-7* and *AtDuf724-10* was mainly detected in the vascular bundle. It is speculated that the DUF724 gene also has similar tissue specificity in wheat [15]. They are actively expressed in plant tissues, which means that they may play important roles in plant growth and development. Previously, ten DUF724 family members were identified in Arabidopsis and five were identified in rice [15]. In comparison, it seems that the number of *DUF724* genes in wheat was higher than that in Arabidopsis and rice simultaneously, which may be due to wheat being an allohexaploid plant with a high retention rate of homomers [16].

Based on the phylogenetic relationships, wheat *TaDUF724s* were divided into two subgroups, Group I and Group II. Group I contained amino acid sequences of wheat, rice and Arabidopsis, while Group II contained only amino acid sequences of wheat and rice (Figure 1). Arabidopsis is dicotyledon, while wheat and rice are monocotyledon. Therefore, the DUF724 domain exists in both monocotyledon and dicotyledon, and can show a certain degree of discrimination.

The generation of new genes mainly depends on gene replication, which plays an extremely important role in the evolution and expansion of plant gene families [17]. In this study, fourteen *TaDUF724* genes were unevenly distributed on 11 chromosomes (Figure 2A), and most *TaDUF724s* showed direct or indirect collinearity with high genetic similarity (Figure 2B). This suggests that *TaDUF724s* may have been added or lost in the evolutionary process. The nonsynonymous substitution rate (Ka), the synonymous substitution rate (Ks) and their ratio (Ka/Ks; sometimes termed dN/dS) are commonly used to reveal the direction of evolution and its selective strength on a coding sequence. Ka/Ks > 1 indicates a positive selection, Ka/Ks < 1 indicates a negative selection and Ka/Ks = 1 indicates a neutral evolution [18]. In this study, Ka/Ks values of homologous gene pairs between each two of *T. aestivum*, *T. turgidum* and *T. urartu* were calculated; the results showed that the Ka/Ks values were all less than one (Figure 2C), indicating that *TaDUF724s* were subjected to the pressure of negative selection in the process of evolution.

Exon–intron analysis is not only an important feature of gene evolution, but also an important clue of gene functional differentiation [19]. When the gene family evolves, the coding region of a gene may be differentiated by gene replication. Therefore, the types and amounts of amino acids may change, which causes the functional differentiation of genes to adapt to different growth conditions [20]. Similarly, this study found that the length and number of exons were different among two groups. The exons of the genes in Group II were longer, but the exons in Group I were more numerous and complex (Figure 3B). Genes with few or no introns are thought to be rapidly activated in response to various stresses [21]. The most common are heat shock protein genes and seed storage protein genes, which have a small number of introns. For example, most *Malus pumila HSP20* (*Heat shock protein 20*) genes were rapidly induced after 4 h of heat stress, which may support a rapid response [22]. In *Cicer arietinum*, the genes encoding SSP have undergone intron loss during evolution in order to be more efficient in transcription [23]. Similarly, in this study, the introns of Group II genes were also missing (or only one). This may be one of the reasons for the rapid induction of *TaDUF724s*. This result explains that the 3D protein model varies greatly between two groups, but the similarity of members within the same group is high (Figure 4). Proteins with similar structures often have similar functions [24]. Therefore, it is speculated that these two sets of genes may have different functions during growth and development. Using conservative Motif analysis, it was found that the number and type of Motif in different groups was very different, and the closer the relationship, the smaller the difference. Proteins that contain highly consistent amino acid sequences, especially in functional domains, often have similar biological functions [25]. According to function annotation of conserved domains, Motif1 constituted the DUF724 domain, Motif6 constituted the Agenet domain, Motif7 and Motif8 constituted the Tudor_SF domain and other Motifs did not match known key functional domains. Motif1, which overlapped with the DUF724 domain, was present in both Group I and II members. Interestingly, Motif1 was the only known function Motif detected in Group II members, while the known function Motifs of Motif6, Motif7 and Motif8 were also contained by Group I members (Figure 3C). DUF724 domains often co-occur with plant Agenet/Tudor domains, which led to functional differentiation of the DUF724 protein [12]. The result further suggests that the Group I members have possibly evolved complex functions. According to this phenomenon, it can be inferred that, with the evolution of *TaDUF724s*, a part of a Motif may be missing or added, thus leading to larger differentiation of gene function.

In order to improve our basic understanding of gene regulation, it is essential to study promoters that primarily regulate gene expression at the transcriptional level [26]. In this study, we identified different functional types of *cis*-regulatory elements in the promoter region of *TaDUF724* genes (Figure 5). These regulatory elements are further divided into light response elements, stress response elements and growth and development elements. It is speculated that these elements play an important role in wheat growth regulation and stress response.

At least seven DUF724 proteins in Arabidopsis are localized in the nucleus, and proteins containing the DUF724 domain may play a novel role in RNA transport [15]. In this study, ten *TaDUF724* genes were located in the nucleus (Table 1). Combined with the *cis*-acting element analysis results of the DUF724 gene family in wheat, common *cis*-acting elements in promoter and enhancer regions (CAAT-box) and core promoter elements around −30 at the start of transcription (TATA-box) were widely distributed and in large numbers [27]. Therefore, it can be speculated that DUF724 family genes enhance their own transcription, and provide a certain theoretical basis for speculations that they play a new role in RNA transport.

Abiotic stresses, such as drought, salt and extreme temperature, limit crop acreage and affect crop quality and yield [28]. The gene encoding the DUF724 domain in algae cells was up-regulated about 66-fold under acidizing conditions [29]; the overexpression of *AtDUF4* containing the DUF724 conserved domain led to the enlargement of plant organs [30]. In this study, we also analyzed the expression patterns of wheat DUF724 gene family members (Figure 6), and the results show that *TaDUF724s* is involved in tissue and organ development, especially seed, root, leaf, stem, spike and grain development. Combined with previous studies of DUF724 in *Arabidopsis thaliana*, most members of this gene family have high expression levels in flowers and seedlings. *AtDuf724-5* was expressed in different organs, with the highest expression level in seedlings, leaves and flowers. *AtDuf724-7* is also expressed in many different tissues, but expression levels are highest in seedlings, roots, leaves and flowers [15]. These results suggest that the DUF724 gene family plays an important role in wheat growth and development. In addition to the function of *TaDUF724s* in wheat growth and development, RT-qPCR analyses have demonstrated that the transcript level of many *TaDUF724s* is elevated under different abiotic and biotic stress conditions such as drought (PEG 6000), salt (NaCl), *F. graminearum* and *P. striiformis*. In this study, the Yangmai 20, a wheat cultivar susceptible to stripe rust, was selected to performed the responsive patterns of *TaDUF724* to rust disease. However, the induced expression pattern of *TaDUF724s* is resistant cultivar is unknown. It will be interesting to compare the expression patterns of *TaDUF724s* between resistant and susceptible cultivars and, which will provide valuable clues for further investigating their biological roles in disease resistance in the future.

MicroRNA (miRNA), a broad class of small non-coding endogenous RNAs with a length of about 21 nucleotides (nt), are key regulatory factors of plant gene expression and play important negative regulatory roles at both the transcriptional and post-transcriptional levels of plants [31]. They control target genes at the post-transcriptional or translational levels of protein synthesis and provide extensive regulation of growth, development and adaptive responses to abiotic stresses [32]. For example, under drought stress, miR398 expression levels in wheat leaves and roots were up-regulated [33]. miR160 targets auxin response factor *ARFs* to influence ABA and auxin signaling under drought stress [34]. In wheat, tae-miR164, tae-miR9661-5p, tae-miR5384-5p and tae-miR9676-5p target *TaGH9-9*, *TaGH9-16*, *TaGH9-15* and *TaGH9-19* to participate in the transformation of pollen fertility [35]. In this study, 12 miRNAs were identified to target 12 *TaDUF724* genes via cleavage and translation inhibition (Figure 7). And tae-miR9676-5p was also found to target *TaDUF724-10* and *TaDUF724-11* in the *TaDUF724* gene family. It is speculated that miRNA and *TaDUF724s* form a regulatory network of miRNAs–target genes to participate in pollen transformation of wheat. Interestingly, *TaDUF724s* has similar *cis*-elements but relatively different expression profiles, such as *TaDUF724-8*, *TaDUF724-9* and *TaDUF724-10*. Further studies revealed that *TaDUF724-8* and *TaDUF724-9* are targeted by tae-miR9653a-3p, while *TaDUF724-10* is targeted by tae-miR1120b-3p. According to previous findings, the miRNAs tae-miR1120b-3p were identified as regulators of the helicase gene Wheat_newGene_324372 in wheat. circ.5B18901457 might regulate the expression of the helicase gene mediated by tae-miR1120b-3p [36]. Studies on Arabidopsis have shown that CCCH zinc finger proteins mediate the shuttling of RNA between the nucleus and cytoplasm by binding RNA, which is regulated by the wheat-specific miRNA tae-miR9653a-3p [37,38]. This may be the reason why *TaDUF724s* have similar *cis*-elements but show very different expression profiles. It is possible that nuclear miRNAs bind enhancers and alter the chromatin state of enhancers, thereby act the transcriptional expression of genes [39].

## 4. Materials and Methods

### 4.1. Identification of DUF724 Gene Family Members in Wheat

The wheat reference genome (IWGSC v2.1) was downloaded from International Wheat Genome Sequencing Consortium website (https://wheat-urgi.versailles.inra.fr/Seq-Repository/Assemblies/, accessed on 12 September 2023) [40]. The Hidden Markov Model of DUF724 domain (PF05266) was downloaded from the Pfam database (v35.0, http://Pfam.xfam.org/, accessed on 12 September 2023) [41], and ten AtDUF724 and five OsDUF724 protein sequences were collected from Arabidopsis and rice [15]. They were merged and used as reference sequences to perform BLASTp (e-value < 10^−5^). The Pfam database was used to further screen out the proteins containing the conserved domain of DUF724, which were members of the wheat DUF724 family [42].

### 4.2. Multiple Sequence Alignment and Phylogenetic Trees

ClustalW2 was used for multiple sequence alignment of DUF724 proteins in wheat, Arabidopsis and rice [43]. The phylogenetic tree was constructed through the neighbor-joining method with 1000 replicated bootstraps. And using iTOL tool (http://ITOL.embl.de/, accessed on 12 September 2023) to infer the phylogenetic relationships of TaDUF724s [44]. According to DUF724 family members’ evolutionary relationships to classify TaDUF724s [45].

### 4.3. Chromosome Localization, Collinearity and Ka/Ks Analysis

The location information of *TaDUF724s* were obtained from the GFF3 annotation file, and the chromosome distribution map was drawn using MapInspect software (32 bit version) [8]. TBtools was used to analyze the relationship between *TaDUF724s* tandem repetition and fragment repetition, and drew the homologous gene map [45]. The homologous gene pairs of DUF724 between each two of *T. aestivum*, *T. turgidum* and *T. urartu* were calculated via BLASTn search, and the Ka (non-synonymous replacement rate), Ks (synonymous replacement rate) and Ka/Ks of gene pairs were calculated using TBtools [46].

### 4.4. Analysis of Protein Conserved Motifs and Gene Structure

According to the GFF3 gene structure annotation information, TBtools was used to map the gene exon/intron structure of *TaDUF724s*. The conserved Motifs of wheat DUF724 protein were predicted using MEME online website (version 5.5.1, http://meme-suite.org/tools/meme/, accessed on 12 September 2023) [47]. The maximum number of Motifs was 10, and the width of Motifs ranged from 6 to 50 aa. The results were analyzed with TBtools and the protein Motif structure map was drawn [47].

### 4.5. Characteristic Analysis of TaDUF724 Protein

The online website ExPASyServer10 (https://prosite.expasy.org/PS50011/, accessed on 12 September 2023) was used to analyze the basic characteristics of TaDUF724, such as protein length (Len), molecular weight (MW), isoelectric point (pI) and total hydrophily (GRAVY) [48]. The website SignalP5.0 (http://www.cbs.dtu.dk/services/SignalP/, accessed on 12 September 2023) was used to predict signal peptides and Plant-mPLoc (http://www.csbio.sjtu.edu.cn/bioinf/, accessed on 12 September 2023) was used to predict subcellular localization [8]. Protein secondary structure prediction was performed via SOPMA online website (http://npsa-pbil.ibcp.fr/cgi-bin/npsa_automat.pl?page=npsa_sopma.html/, accessed on 12 September 2023). The 3D protein model was constructed through SWISS—MODEL online website (https://www.swissmodel.expasy.org/, accessed on 12 September 2023).

### 4.6. Promoter Cis-Acting Element Analysis

TBtools was used to retrieve the CDS sequence of 1500 bp upstream of *TaDUF724* genes, and PlantCARE website (http://bioinformatics.psb.ugent.be/webtools/plantcare/html/, accessed on 12 September 2023) was used to identify the *cis*-element in the promoter region [49]. The R software package (v4.2.1) “pheatmap” was used to display the analysis results.

### 4.7. Expression Pattern Analysis of TaDUF724s

Wheat transcriptome data were downloaded from the NCBI Short Read Archive database (PRJEB25639, PRJEB23056, PRJNA436817, SRP133837, PRJEB25640 and PRJEB25593), then the fastqc and trim_galore were used for quality control and filtering, respectively. The reads were mapped to the wheat reference genome using hisat2 [50]. Then, the expression levels of *TaDUF724s* were calculated and normalized using Cufflinks (represented by TPM, transcripts per kilobase of exon model per million mapped reads) [51]. The R software package (v4.2.1) “pheatmap” was used to draw heat maps to show the expression patterns of *TaDUF724* genes under different conditions [52].

### 4.8. Prediction of Targeting Relationship between miRNA and TaDUF724s

MicroRNA maturation sequences of wheat were collected from publicly available reports [53], and miRNA sequences and CDS sequences of *TaDUF724s* were submitted to psRNA Target (https://www.zhaolab.org/psRNATarget/analysis?function=3/, accessed on 12 September 2023). The targeting relationship between miRNA and *TaDUF724s* was analyzed, and the R software packages (v4.2.1) “ggplot2” and “ggalluvial” were used to draw the targeting relationship between miRNAs and *TaDUF724s*.

### 4.9. Wheat Materials and Handling

Yangmai 20 was selected as the wheat seed variety, which is moderately susceptible to *Fusarium* head blight and highly susceptible to stripe rust [45]. Surface sterilization was first performed with 1% sodium hypochlorite solution, then rinsed with ddH_2_O and incubated in a Petri dish at 25 °C for 2 days [54]. The germinated seeds were then cultured in a 1/4 strength Hoagland solution, in a photoperiod of 16/8 h (day and night) at 25 °C [45]; the intensity increased to 1/2 after 3 days. *F. graminis* (PH-1) spores were obtained according to the method studied [55]. When the seedlings were at one leaf stage, 10 uL spore suspension (5 × 10^5^/mL^−1^) was absorbed and dropped on the leaves. We kept them hydrated with a wet paper towel and cultured at 25 °C and 65% relative humidity. Uninoculated wheat seedlings were served as controls. Wheat leaves were taken at 6, 12, 24, 48, 72 and 96 h after inoculation. According to the method of Zhan et al. to obtain stripe rust (CYR32) spore, at the two-leaf stage, we sprayed fresh rust spore suspension (4 mg/mL) and avoided light and moisture for 24 h. Growth conditions were set as 16 h/8 h (day/night) photoperiod at 18 °C. Uninoculated wheat was the control, and leaves were taken at 6, 12, 24 and 48 h after inoculation [56]. For salt or drought treatment, when wheat grew into one leaf and one heart stage, seedlings were treated with 400 mM NaCl (salt) or 20% PEG 6000 (drought), respectively. The leaves were then collected at 2, 6, 12, 24, 48 and 72 h after treatment [57]. Each sample includes three biological replications.

### 4.10. RNA Extraction and RT-qPCR

Total RNA was extracted using TRizol reagent (GenStar, Beijing, China) and DNA was removed with DNaseI (Vazyme, Nanjing, China) enzyme. Reverse transcription of RNA into complementary DNA (cDNA) was performed using RevertAid reverse transcriptase (Vazyme, Nanjing, China), the cDNA was diluted with enzyme-free water [58]. *TaDUF724* gene-specific primers were designed using Primer Premier 5 software (Table 3). The RT-qPCR analysis was performed on the CFX 96 Real-Time PCR System (BioRad, Hercules, CA, USA) using ChamQ SYBR qPCR Master Mix (Vazyme, Nanjing, China), according to the manufacturer’s instructions [59]. The quantitative real-time polymerase chain reaction consisted of 2 × SYBR Premix Extaq 10 μL forward and reverse primers, 0.4 μL each. cDNA was diluted to 2 μL and ddH_2_O was composed of 7.2 μL. The program contains three steps: 95 °C pre-denaturation for 30 s (step 1); 95 °C denaturation for 5 s (step 2); 60 °C primer annealing, extension and fluorescence signal collection for 30 s (step 3). The next 40 cycles began at step 2. The technique was repeated 3 times for each cDNA. The relative expression level was measured by 2^−ΔΔCt^ [60]. ADP-ribosylation factor Ta2291, whose expression was stable under various conditions, was used as an internal reference gene for RT-qPCR analysis. Three biological replicates and three technical replicates were performed for each treatment.

## 5. Conclusions

In summary, a total of 14 *TaDUF724* genes were identified in wheat and divided into two subgroups (Group I, Group II). Group I contained 11 *TaDUF724* genes and Group II contained 3 *TaDUF724* genes. They were unevenly distributed across the 11 chromosomes. Fragment replication and negative selection is the main way the *TaDUF724* family is extended. TaDUF724 is a hydrophilic protein, and the main spatial structure is *α*-helix and random coil; it is distributed in the nucleus and chloroplast. Meanwhile, *Cis*-elements associated with growth and development were detected in large numbers in the promoter region of the *TaDUF724* gene family. The expression patterns indicated that the *TaDUF724* gene family plays an important role in growth and development and stress response. MiRNAs and *TaDUF724s* form a complex network to regulate expression patterns in different environments. RT-qPCR confirmed that the *TaDUF724* gene family was widely involved in biotic and abiotic stress.

## Figures and Tables

**Figure 1 ijms-24-14248-f001:**
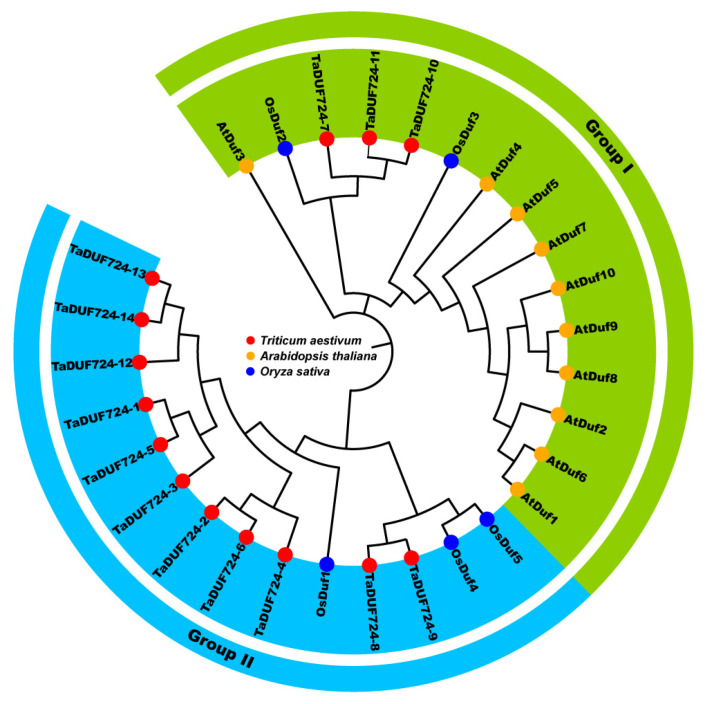
Phylogenetic tree of the DUF724 proteins from three species. Ten *Arabidopsis thaliana*, five *Oryza sativa* and fourteen *Triticum aestivum*. Different-colored circles represent different species.

**Figure 2 ijms-24-14248-f002:**
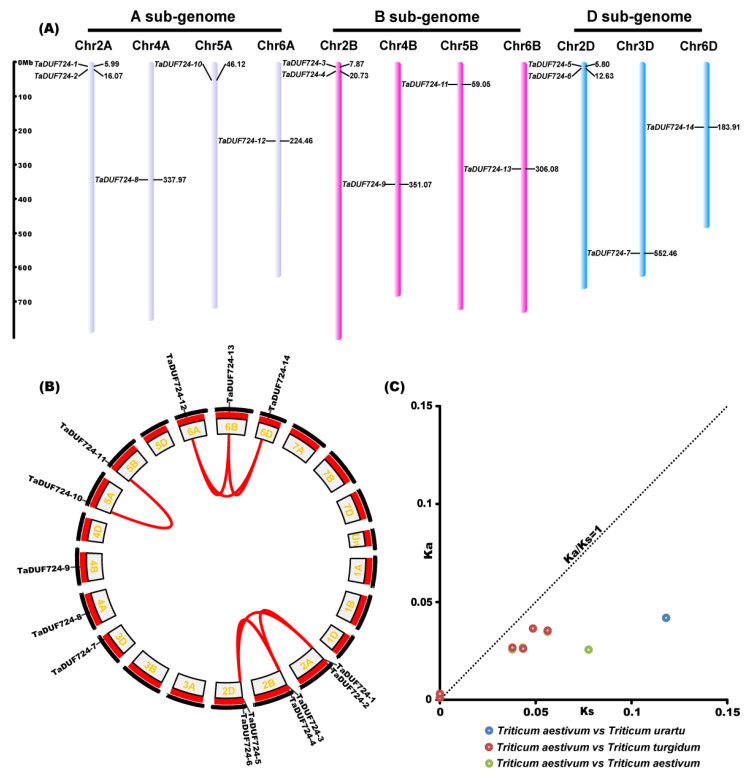
Chromosome localization, collinearity analysis and Ka/Ks analysis of *TaDUF724* genes. (**A**) Chromosome distribution of *TaDUF724s*. Chr stands for chromosome. The scale on the left represents the physical length of chromosome (Mbp). (**B**) Collinearity analysis diagram of *TaDUF724s*. (**C**) Ka and Ks scatter diagram of *DUF724* homolog gene pairs among wheat and different species.

**Figure 3 ijms-24-14248-f003:**
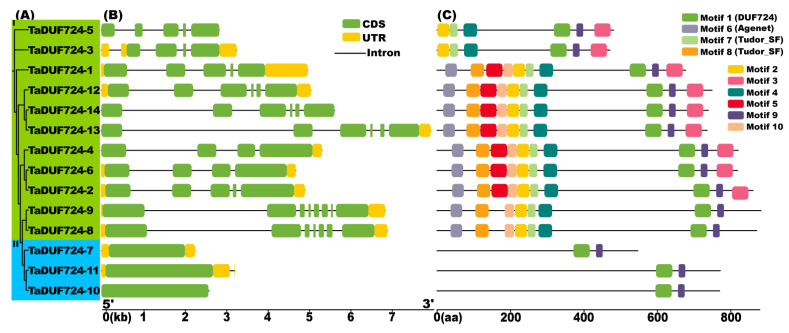
Phylogenetic tree, gene structure and conserved Motif of TaDUF724s. (**A**) Phylogenetic tree of TaDUF724s. (**B**) Gene structure of *TaDUF724s*, green represents CDS, yellow represents UTR and black lines represent introns. (**C**) Motif distribution of TaDUF724s, different colors represent different Motifs.

**Figure 4 ijms-24-14248-f004:**
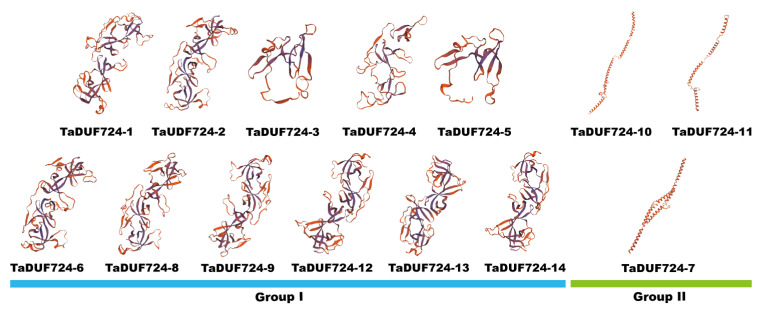
Predicted 3D models of TaDUF724s. Models were generated by using SWISS-MODEL.

**Figure 5 ijms-24-14248-f005:**
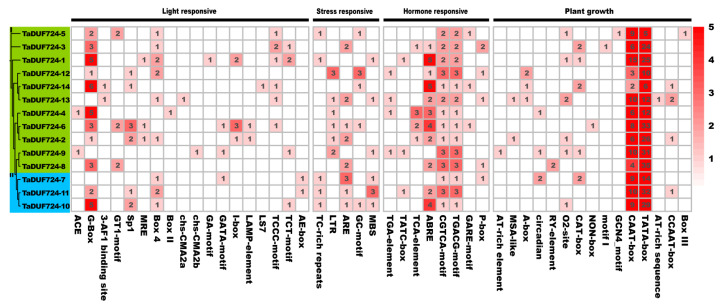
Statistical analysis of *cis*-acting elements of *TaDUF724* promoters. The color and number of grids represent the number of *cis*-acting elements in the corresponding *TaDUF724s*.

**Figure 6 ijms-24-14248-f006:**
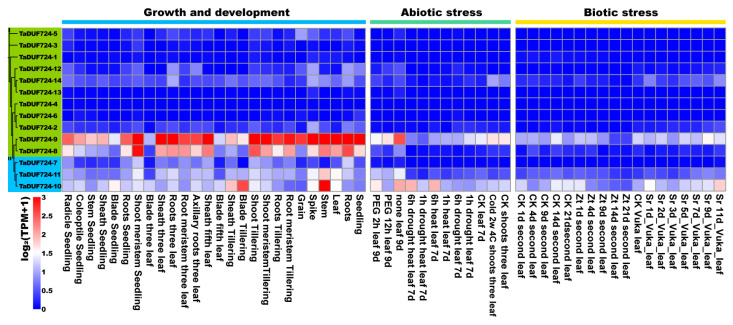
Expression patterns of *TaDUF724s* during growth and development, biological and abiotic stress treatments. The log2 (transcripts per kilobase of exon model per million mapped reads + 1) values were used to draw the heatmap. Zt stands for *Zymoseptoria tritici* and Sr stands for stripe rust.

**Figure 7 ijms-24-14248-f007:**
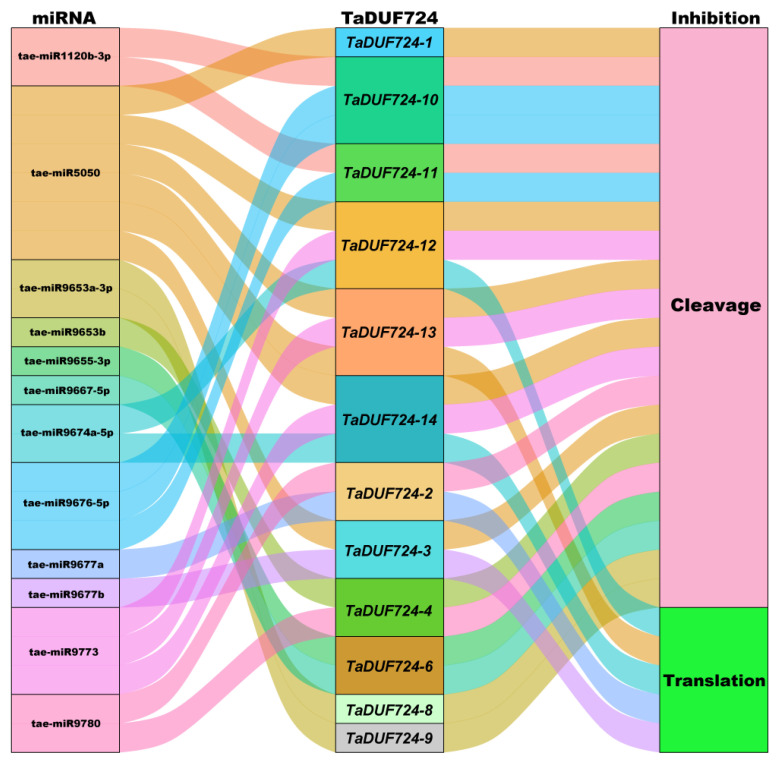
Sankey diagram for the relationships of miRNA targeting to *TaDUF724* transcripts. The three columns represent miRNA, *TaDUF724s* and inhibition effect.

**Figure 8 ijms-24-14248-f008:**
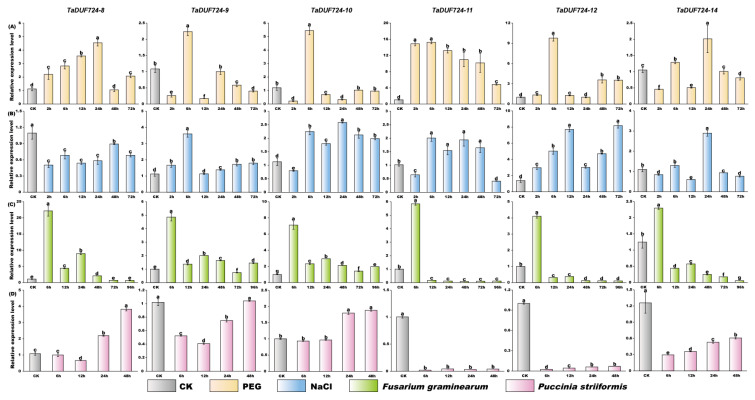
RT-qPCR expression analysis of six selected *TaDUF724* genes in response to (**A**) PEG, (**B**) NaCl, (**C**) *F. graminearum* and (**D**) *P. striiformis*. Different colors represent different treatments. Vertical bars indicate the standard error of the mean, and different letters are significantly different (*t*-test, *p* < 0.05).

**Table 1 ijms-24-14248-t001:** Protein characterization of TaDUF724s.

Name	Gene ID	Len	MW	Pi	Ins	Sta	GRAVY	Sig	Sub
TaDUF724-1	*TraesCS2A03G0023500.1*	752	82.4	5.8	65	not	−0.436	no	Nuc
TaDUF724-2	*TraesCS2A03G0056500.1*	956	106	5	51.4	not	−0.465	no	Nuc
TaDUF724-3	*TraesCS2B03G0029300.1*	523	57.8	6.1	63	not	−0.458	no	Nuc
TaDUF724-4	*TraesCS2B03G0083400.1*	911	101	4.7	54	not	−0.451	no	Nuc
TaDUF724-5	*TraesCS2D03G0020600.1*	534	58.5	6.2	58.3	not	−0.354	no	Chl. Nuc
TaDUF724-6	*TraesCS2D03G0057200.1*	909	100.5	4.9	50.3	not	−0.464	no	Nuc
TaDUF724-7	*TraesCS3D03G0969100.1*	607	69.6	5.1	55.8	not	−0.804	no	Nuc
TaDUF724-8	*TraesCS4A03G0406500.1*	967	107.5	7.6	58	not	−0.62	no	Nuc
TaDUF724-9	*TraesCS4B03G0462800.3*	980	108.9	7.6	55.4	not	−0.61	no	Chl. Nuc
TaDUF724-10	*TraesCS5A03G0123500.1*	855	96.4	5.8	58.3	not	−0.529	no	Chl. Nuc
TaDUF724-11	*TraesCS5B03G0134600.1*	857	96.9	5.5	58.8	not	−0.536	no	Chl
TaDUF724-12	*TraesCS6A03G0468800.1*	832	92.1	6.1	55.8	not	−0.524	no	Chl
TaDUF724-13	*TraesCS6B03G0586100.1*	817	90.6	6	56.7	not	−0.478	no	Chl
TaDUF724-14	*TraesCS6D03G0391800.1*	821	90.9	6.4	56.8	not	−0.481	no	Chl

Len: length of amino acid (aa); MW: molecular weight (kDa); Pi: isoelectric point; Ins: instability index; Sta: stability; GRAVY: grand average of hydropathicity; Sig: signal peptide; Sub: subcellular localization; Nuc: nucleus; Chl: chloroplast.

**Table 2 ijms-24-14248-t002:** Protein secondary structure analysis of TaDUF724s.

Name	Alpha Helix (%)	Extended Strand (%)	Beta Turn (%)	Random Coil (%)
TaDUF724-1	36.44	11.97	7.18	44.41
TaDUF724-2	34.52	13.81	5.44	46.23
TaDUF724-3	44.36	9.37	7.27	39.01
TaDUF724-4	33.37	16.03	6.70	43.91
TaDUF724-5	43.63	9.18	6.37	40.82
TaDUF724-6	32.45	15.95	7.04	44.55
TaDUF724-7	44.48	10.71	5.60	39.21
TaDUF724-8	35.68	12.62	5.27	46.43
TaDUF724-9	35.31	14.29	6.53	43.88
TaDUF724-10	44.68	10.06	5.03	40.23
TaDUF724-11	44.81	9.22	5.37	40.61
TaDUF724-12	33.77	11.42	5.17	49.64
TaDUF724-13	35.25	12.36	5.02	47.37
TaDUF724-14	35.57	12.79	5.24	46.41

**Table 3 ijms-24-14248-t003:** Specific primers of *TaDUF724s*.

Gene Name	Forward Primer	Reverse Primer
*Ta2291*	GCTCTCCAACAACATTGCCAAC	GCTTCTGCCTGTCACATACGC
*TaDUF724-8*	CCAAGGTCGCCATTTCTGTC	GGGGAACCACGAGTAGCCATA
*TaDUF724-9*	CAAGCATTGGAGGATTGTAAGTC	TTTTTGTGGCTGGAGATTGTG
*TaDUF724-10*	ATGGTTGAATGGGCACTTTG	ACATTTATCACCTGCCGTATCTG
*TaDUF724-11*	AAGAGGGTGGAAGAAAGCAAGA	TTGTTCACCAACCCTCTGTAATG
*TaDUF724-12*	TTGCTTGAGGCAGATGGTTT	TTCTTGGTCCTTGGTTTCTTTG
*TaDUF724-14*	GTATTAAGCGGTCAGGGTAAGC	CCTAGAGCCACTGCTGTTTGA

## Data Availability

All datasets supporting the conclusions of this article are included within the article. The genome data and sequences and expression profiles of *TaDUF724* genes used in the current study are available in the Wheat Whole Genome Database (https://wheat-urgi.versailles.inra.fr/Seq-Repository/Assemblies/, accessed on 12 September 2023). The datasets generated and analyzed during the current study are available from the corresponding author upon reasonable request.

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
