# Peer review of "Genome-Wide Identification, Characterization and Expression Analysis of the *TaDUF724* Gene Family in Wheat (*Triticum aestivum*)"

_ijms, 2023, doi:10.3390/ijms241814248_

Round 1

Reviewer 1 Report

The subject of the study is interesting and topical, with scientific and practical importance.

The introduction is presented correctly, in accordance with the subject. Numerous scientific articles, in concordance to the topic of the study, were consulted.

Methodology of the study was clearly presented, and appropriate to the proposed objectives.

The obtained results are important and have been analyzed and interpreted correctly, in accordance with the current methodology. It is recommended that some issues be reviewed.

The discussions are appropriate, in the context of the results, and was conducted compared to other studies in the field.

The scientific literature, to which the reporting was made, is recent and representative in the field. 

The following aspects are brought to the attention of the authors:

What about the biological replicas in the qPCR analysis?

The data used in the analyses should be better described, so far I can see that there is information from which database they are, but I am missing accession numbers, more precise bearings on these data.

Reviewer 2 Report

The authors have done a good job in presenting the genome-wide identification and characterization of wheat DUF724 gene family. Over all the manuscript looks ok. Please see following comments and also the attached document:

1. More emphasis should be given to why the TaDUF724 could potentially be important, like if possible add more references in the introduction section.

2. The names of the genes are bit confusing when writing that 14 genes across 11 chromosomes instead either it should be 14 genes in whole wheat genome or 14 genes on 11 chromosomes. Also It might be a good idea to add chr. number with each gene to make it easier to read and understand simultaneously.

3. The discussion part looks like an extension of results, the authors are advised to add more context especially link previous studies of the gene family and their correlation in wheat and also in other study.

4. Authors can also add the changes in expression of the rice and arabidopsis genes used in the current study and if the previous study validates the current information generated about the TaDUF724 gene family.

5. looking at Figure 5 there is hard to tell any difference amongst all the genes in both the groups (inter or intra). However, in Fig. 6 clearly few genes like TaDUF8, TaDUF9 and TaDUF10 have relatively different expression profile. Authors comments on why is it like this in the discussion part should be included.

6. The auhtors should write briefly why the cultivar Yangmai 20  was selected for the study, Also was CYR23 virulent or avirulent on Yangmai 20? It is important because Pst can be qualitatively controlled under such case it would be interesting to see if the expression of TaDUF724 changes in resistant or susceptible line in the presence/absence of the resistance gene, author should comment on that.

7. There were several edits and mistakes in the manuscript, Please proofread the manuscript for mistakes and English language before final submission.

8. Further comments and corrections are in the attached pdf file.

The manuscript should be proofread before final submission for both english language and other mistakes.

Reviewer 3 Report

Manuscript "Genome-wide identification, characterization and expression analysis of TaDUF724 gene family in wheat (Triticum aestivum)" by Yuan et al. is well-elaborated research that investigates the Durf724 gene family in bread wheat. The study involves both in silico bioinformatical and experimental qPCR research, which makes the results solid and valid. 

However, there are certain issues that should be addressed or corrected.

1. What was the biological replication of the RT-qPCR experiment?

2. The methodology of P. striiformis inoculation is absent and should be added.

3. 422-423 "At the same time, wheat was treated with 400 mM NaCl (salt)": the meaning is not clear; do you mean you treated plants with PEG, NaCl, and Fusarium graminis simultaneously? 

4. I suggest that you add the search for long non-coding RNAs that target the identified TaDUF724s sequences as lncRNA has been recently recognized as one of the major drivers of gene expression regulation. 

5. Please include in the discussion the absence (or just one) of introns in Group II TaDUF724. The absence of introns occurs, for example, in seed storage proteins and heat shock proteins, i.e., it can be explained by their high expression in the necessary moment: no introns, no splicing; therefore, less time is required for processing.

6. In the abstract, the sentence "DUF724s play crucial roles in plants" is too short and should be extended. At least, DUF should be deciphered, and its importance should be briefly explained.

7. Huang et al. (reference [40]) are not developers of the PlantCARE website; reference [40] should be replaced with Lescot et al. (2002) Lescot M, Déhais P, Thijs G, Marchal K, Moreau Y, Van de Peer Y, Rouzé P, Rombauts S. PlantCARE, a database of plant cis-acting regulatory elements and a portal to tools for in silico analysis of promoter sequences. Nucleic Acids Research. 2002;30:325–327. doi: 10.1093/nar/30.1.325.

The paper can be accepted for publication after minor revision.

Kind regards,

Reviewer

Wording and incorrect formatting should be corrected

"TaDUF724" (where it is used as a gene designation) should be in italics (e.g. lines 16, 18,  139, 314, 373, 400, 428)

114 "Triticum Urartu" should be "Triticum urartu"

402 "collected from reported literatures" should be rephrased (e.g. "from publicly available reports").

419-420 "Culture 420 16/8 hours (day/night) photoperiod under 18°C light": the sentence is incomplete.

434-435 "The process is as follows :95°C" should be "The process is as follows: 95°C"

440 "Table 3. specific" should be "Table 3. Specific'

Latin and gene names in References should be initalics (e.g. Triticum aestivum in 486, 505, 559, 566, 573,  Arabidopsis thaliana in 502, Pennisetum glaucum in 508, TaDi19A in 520, Symbiochlorum hainanensis in 523, TaLIM8‐4D in 550, Alternanthera philoxeroides in 581)

Please translate reference 26 into English.
